# The Father’s Microbiome: A Hidden Contributor to Fetal and Long-Term Child Health

**DOI:** 10.3390/biology14081002

**Published:** 2025-08-05

**Authors:** Enrica Zambella, Annalisa Inversetti, Silvia Salerno, Martin Müller, Nicoletta Di Simone

**Affiliations:** 1Department of Biomedical Sciences, Humanitas University, 20090 Pieve Emanuele, Italy; annalisa.inversetti@hunimed.eu (A.I.); silvia.salerno@st.hunimed.eu (S.S.); nicoletta.disimone@hunimed.eu (N.D.S.); 2IRCCS Humanitas Research Hospital, 20089 Milan, Italy; 3Department for Gynecology and Obstetrics, Lindenhofspital AG, 3001 Bern, Switzerland; martin.mueller@lindenhofgruppe.ch; 4Institute of Biochemistry and Molecular Medicine, University of Bern, 3012 Bern, Switzerland

**Keywords:** male microbiome, seminal microbiome, sperm epigenetics, paternal programming, offspring health, paternal epigenetic inheritance, preconception health, intergenerational metabolic programming, microbial dysbiosis

## Abstract

The father’s health at the time of conception can influence the long-term well-being of his offspring by modulating the development of the infant gut microbiome. While the maternal microbial contribution is most prominent around birth, the paternal input seems to be stable and increasingly evident as the child grows. Emerging evidence suggests that exposure to environmental toxins, dietary habits, and hormonal balance may influence both sperm quality and the composition of the gut and seminal microbiomes. Recent studies have shown that disruptions in these microbiomes can modify the testicular environment, including metabolite profiles and hormonal levels, and affect chromatin states, small RNAs, and macromolecules within sperm, ultimately influencing fertility and offspring development. Despite these promising findings, most of the current evidence derives from animal models, and there is a significant lack of human studies. A deeper understanding of how the paternal microbiota contributes to fetal and childhood health could pave the way for new preventive strategies, such as preconception dietary or probiotic interventions.

## 1. Introduction

The microbiota refers to the entire community of microorganisms, including bacteria, viruses, fungi, archaea, and protozoa, that inhabit various anatomical sites and exert complex influences on human health and disease. Over the past two decades, growing interest has emerged regarding the role of the microbiota in reproductive health, in both female and male patients. Advances in next-generation sequencing (NGS) have demonstrated that human semen is not sterile but harbors a specific microbial community [1]. Most available evidence focuses on the seminal microbiota in infertile patients, aiming to associate dysbiosis with altered semen parameters [1]. Due to the increasing evidence on the role of seminal fluid in implantation and placental development [2], seminal microbiota may also have an effect on reproductive outcomes and fetal health. Moreover, due to the interactions among gut microbiota and other organs, it has been shown that androgens can remodel the gut microbiota through complex pathways, while gut microbes can, in turn, regulate androgen production and metabolism [3,4]. Similarly, the oral microbiota may influence sperm’s quality, by modulating sperm motility, and it may metabolize and influence systemic hormonal levels [5,6]. These findings suggest that the interplay between male microbiota, hormonal regulation, and reproductive health is modulated by intricate mechanisms that remain incompletely understood. Within this context, the role of the paternal microbiota in influencing pregnancy outcomes and fetal health represents a novel and rapidly evolving field of research, complementing the maternal microbiome literature [7,8]. A recent review by Kilama et al. proposed three primary mechanisms through which the seminal microbiome may mediate paternal programming effects: (1) sperm epigenetic modifications during the initial stages of spermatogenesis; (2) interaction between seminal epididymosomes, extracellular vesicles of 50–250 nm in diameter secreted by the epididymal epithelium and composed of proteins, non-coding RNAs, and specific lipids, and bacteria-derived extracellular vesicles (bEVs), mediators of bacteria–host communication containing proteins, lipids, nucleic acids, and metabolites; and (3) the production of signaling molecules in response to paternal exposure to external environmental factors [9]. Supporting this hypothesis, Argaw-Denboba et al. showed that gut microbiota alterations in male mice can increase the risk of offspring presenting with low birth weight, severe growth restriction, and premature mortality [10]. Furthermore, these effects were reversible after microbiome restoration, supporting the hypothesis of a gut–germline axis to explain the interactions between intestinal flora and the sperm epigenome that influence offspring health [10]. Nowadays, available evidence has been collected from animal studies, owing to the ability to tightly control environmental exposures and easier sampling. However, understanding how preconception lifestyle and environmental factors modulate paternal microbiota composition in humans may provide novel insights into epigenetic inheritance and disease risk in offspring. This narrative review aims to summarize the current state of research on the role of paternal microbiota in shaping fetal outcomes, with a particular focus on neurodevelopmental, metabolic, and gastrointestinal health. Investigating the periconceptional modulation of seminal and gut microbiota may enhance our understanding of the mechanisms of male infertility, reproductive health, and offspring outcomes.

## 2. Male Microbiota

In the following subsections of this paragraph, the main evidence on the relation among microbial composition across different sites and male reproductive health are reported. Figure 1 summarizes the most relevant findings for each anatomical district, illustrating the microbial strains identified in various microbiomes and their associations with reproductive and general health conditions.

### 2.1. Gut Microbiota

The gastrointestinal microbiota is among the most extensively studied microbial communities due to its accessibility. Its impact on the male reproductive system has been investigated during the last two decades. Evidence from both human and animal studies demonstrates that androgens can modulate microbial composition, with males typically exhibiting lower microbial diversity compared to females [11,12]. Moreover, several specific microbial taxa, including *Comamonas testosteroni*, *Butyricicoccus desmolans*, *Clostridium cadaveris*, *Propionimicrobium lymphophilum*, *Clostridium scindens*, and *Clostridium innocuum* express steroid-metabolizing enzymes such as 17,20-desmolase, 20β-HSDH, and 5β-reductase, which are involved in androgen biosynthesis and metabolism [13]. The gut microbiota can also influence spermatogenesis through the production of microbial-associated molecular patterns (MAMPs), such as lipopolysaccharide (LPS), lipoproteins, or peptidoglycans. These MAMPs can enter systemic circulation, reach the testes via the testicular artery, and bind to pattern recognition receptors (PRRs), triggering inflammatory responses. This cascade activates the xanthine oxidase system and increases the production of reactive oxygen species (ROS), potentially damaging Leydig cells and disrupting the blood–testis barrier (BTB) [14,15]. In line with this, obese men with hypogonadism have shown elevated levels of lipopolysaccharide-binding protein (LBP) and pro-inflammatory cytokines, which negatively correlate with serum testosterone levels [16]. Based on these findings, Tremellen et al. proposed the GELDING theory, suggesting that gut endotoxins inhibit steroidogenesis in Leydig cells and impair luteinizing hormone (LH) signaling from the pituitary, leading to decreased testosterone production and impaired spermatogenesis [16]. Additionally, the gut microbiota may regulate the hypothalamic–pituitary–gonadal axis. As shown in Figure 1, patients affected by type 2 diabetes mellitus (T2DM) with testosterone deficiency exhibited more severe gut dysbiosis, characterized by increased abundance of *Streptococcus*, which positively correlated with insulin, C-peptide (CRP), and the homeostatic model assessment of insulin resistance (HOMA-IR), and negatively with LH levels [17]. Emerging data also suggest a role of the gut microbiota in modulating sexual function. As discussed in Figure 1, patients with erectile dysfunction (ED) showed significantly higher levels of *Alistipes* and *Clostridium XVIII* [18] and increased abundance of Lachnospiraceae (family), *Lachnospiraceae NC2004 group*, *Oscillibacter*, *Senegalimassilia*, and *Tyzzerella 3*, according to a recent Mendelian randomization (MR) analysis [19]. The current evidence strongly supports a functional gut–testis axis, whereby the intestinal microbiota influences hormonal homeostasis, spermatogenesis, and sexual function. Disruptions in microbial composition, particularly in the context of obesity, metabolic disorders, and inflammation, may contribute to hypogonadism and infertility. These findings underscore the importance of considering gut health in the assessment and treatment of male reproductive disorders. Future research should aim to validate these associations in larger human cohorts and to explore whether targeted modulation of the gut microbiota, through diet, probiotics, or other interventions, could serve as a novel therapeutic strategy to enhance male fertility and offspring health.

### 2.2. Oral Microbiota

The oral microbiota, defined as the community of microorganisms residing in the oral cavity, exhibits low diversity and richness under healthy conditions and varies according to age and sex [20]. A study analyzing the saliva metagenome of 4478 individuals revealed that women harbored higher levels of *Streptococcus*, *Prevotella*, and *Granulicatella*, whereas men showed increased abundance of *Campylobacter A*, *Veillonella*, *Porphyromonas*, and *Oribacterium*, suggesting a role of sex-specific hormones in shaping the oral microbiota [6]. The authors also identified several sex-specific causal relationships between the oral microbiome and serum metabolites. For instance, elevated levels of carnosine and cystathionine in males were causally linked to a decreased abundance of *Fusobacterium periodonticum C*. Moreover, a causal association between androstenedione and *Pauljensenia* was observed exclusively in males at the tongue dorsum site [6]. Gender-specific differences have also been reported in pathological conditions. For example, significantly higher levels of *Atopobium* were found in the salivary microbiome of male patients with glioma [21], while the genera *Pseudomonas* and *Papillibacter* predominated in male patients with stage III and IV periodontitis [22]. Collectively, these findings reinforce the hypothesis that sex hormones influence the composition of the salivary microbiome. Like the gut microbiota, there is increasing evidence that specific periodontal pathogens can metabolize testosterone. It has been demonstrated that *Aggregatibacter actinomycetemcomitans*, *Prevotella intermedia*, and *Porphyromonas gingivalis* can convert testosterone into dihydrotestosterone (DHT), and 4-androstenedione into testosterone and DHT. Additionally, the oral spirochete *Treponema denticola* exhibits 5α-reductase activity, enabling it to metabolize sex hormones in vitro [23]. The influence of the oral microbiota extends beyond the oral cavity and impacts systemic health. *Fusobacterium nucleatum*, one of the most prevalent Gram-negative anaerobic bacteria in human dental plaque, plays a crucial role in activating the host immune system. However, due to its capacity to migrate from the oral cavity to other body sites, it has been implicated in various systemic pathologies, including Lemierre’s syndrome, acute and chronic mastoiditis, otitis media, sinusitis, cardiovascular diseases, Crohn’s disease, and colorectal cancer (Figure 1) [24]. In the context of male reproductive health, periodontitis, an inflammatory disease affecting the tooth-supporting tissues, has been associated with male infertility. Indeed, periodontal inflammation triggers the systemic release of pro-inflammatory cytokines, such as interleukin-6 (IL-6), interleukin-1 beta (IL-1β), and tumor necrosis factor-alpha (TNF-α) [25]. These cytokines may disrupt the hypothalamic–pituitary–gonadal axis, ultimately impairing testosterone secretion. According to a recent meta-analysis of nine studies involving 1386 participants, periodontal disease (including gingivitis and periodontitis) is significantly associated with reduced sperm motility, abnormal morphology, and increased DNA fragmentation [26]. Notably, men with periodontal disease have a doubled risk of ED, possibly due to chronic inflammation that may inhibit nitric oxide signaling and thereby impair erectile function [27,28]. However, to our knowledge, no studies have yet identified specific microbial species directly related to male infertility or sexual dysfunction in males.

### 2.3. Semen Microbiota

Recent research has focused on fathers’ biological contribution to pregnancy. Reproductive outcomes extend beyond the quality of male gametes at conception [2]. Over the past two decades, studies have demonstrated that acellular seminal plasma interacts with the female reproductive tract through bioactive signaling molecules, influencing fertility, fecundity, and offspring health. Initially, seminal fluid interacts with luminal epithelial cells on the endometrial surface, differentially regulating the expression of mRNAs and non-coding microRNAs (miRNAs) [29]. This interaction subsequently induces the local synthesis of a range of cytokines and chemokines, which activate neutrophils and macrophages to clear debris and microbes introduced during coitus, as well as to select sperm most competent for fertilization [30]. Additionally, dendritic cells are stimulated to promote a regulatory T cell response, enhancing endometrial receptivity for embryo implantation. Indeed, exposure to seminal fluid, whether via vaginal intercourse or intravaginal, intracervical, or intrauterine administration, has been associated with a 23% improvement in clinical pregnancy rates [31]. Using in vitro models involving primary ectocervical epithelial cells or the immortalized ectocervical cell line Ect1, transforming growth factor-beta isoforms (TGF-β1, TGF-β2, and TGF-β3), E-series prostaglandins (PGEs), and ligands of Toll-like receptor 4 (TLR4), such as bacterial LPS, have been found to represent the key components of seminal plasma that regulate immune response genes in cervical cells [32]. Lastly, studies in mice showed that pregnancies conceived in the absence of seminal plasma contact exhibited altered offspring development, including abnormal growth trajectories and impaired metabolic function, characterized by increased central adiposity, dysregulated metabolic hormones, reduced glucose tolerance, and hypertension [33]. Building on this body of evidence regarding the role of seminal fluid in reproduction and thanks to advances in sequencing techniques that have established seminal fluid as naturally colonized by microbes, the role of the seminal microbiome has increasingly been investigated. The studies primarily focus on the seminal microbiota of male partners in infertile couples, aiming to assess potential associations between bacterial composition and semen parameters, such as total sperm count, concentration, motility, and morphology. Since infertility may involve the female partner, such cohorts often include fertile men with normal semen parameters who serve as internal controls for comparison with those presenting abnormal spermiograms. Alternatively, the seminal microbiota has also been examined in relation to pathological conditions such as viral infections (e.g., human immunodeficiency virus—HIV—and human papillomavirus) or prostatitis. One of the earliest studies on the seminal microbiota, conducted by Weng et al., identified three distinct seminal bacterial community types: G1 (*Pseudomonas*-predominant), G2 (*Lactobacillus*-predominant), and G3 (*Prevotella*-predominant). The authors reported a positive association between the G1 community type and improved sperm quality, whereas the G3 type was more frequently observed in low-quality semen samples [34]. Baud et al. reported similar findings: *Prevotella* spp. were positively associated with abnormal spermiograms and reduced motility (Figure 1), *Lactobacillus* spp. correlated with normal sperm morphology, and *Staphylococcus* spp. were linked to both normal semen profiles and high total motility [35]. On the other hand, other studies observed a complete absence of *Lactobacillus* and *Prevotella* spp. in semen samples. Indeed, infertile patients showed increased relative abundances of *Aerococcus*, *Rhodocytophaga*, and *Gemella*, while *Collinsella* spp. were more prevalent in normospermic men [36,37]. In conclusion, according to a recent review of the literature, *Bacteroides*, *Prevotella*, *Ureaplasma*, *Corynebacterium*, and *Lactobacillus* could have a role in semen quality, but further investigations are needed [1]. Definitive results regarding the total bacterial load in semen and the abundance of specific bacterial genera or species in infertile men remain elusive. These inconsistencies may be attributed to the small sample sizes of existing studies, the high heterogeneity of study populations, and the limited knowledge of seminal microbiota composition in fertile men [1]. Moreover, genus-level taxonomic resolution may not be sufficient to fully assess the impact of seminal bacteria on sperm parameters [38]. To establish a meaningful correlation between microorganisms and semen quality, future studies should include more homogeneous populations, larger sample sizes, and species-level taxonomic resolution. As a result, the role of the semen microbiome in male factor infertility is still a subject of ongoing debate. The concept of a “complementary semino-vaginal microbiota” may offer additional insight into the effects of seminal fluid on fertility through the potential transfer of microorganisms from the male to the female reproductive tract. A recent study on infertile couples found that the seminal microbiome closely resembled the vaginal microbiota of the female partner, with *Prevotella* being the most abundant genus in both samples [39]. Once the seminal microbiota is introduced into the female reproductive tract, PRRs expressed by immune cells in the uterine microenvironment may trigger a cascade of immunological responses and cytokine production essential for embryo implantation [40]. Additionally, bEV-associated molecules in seminal fluid can activate both surface and intracellular receptors on endometrial cells, initiating innate immune signaling and promoting immunomodulation through inflammatory pathways [9].

### 2.4. Male Genital Microbiota and Technical Issues

The female genital tract has been extensively studied, whereas the metagenomic characterization of the male genital tract remains in its preliminary stages. Due to the anatomical structure of the male reproductive system, sampling is highly invasive. Consequently, most available data have been derived from pathological conditions, such as prostate cancer, or testicular biopsies performed during infertility assessments. For this reason, the male genital tract microbiome has been indirectly investigated through the semen microbiota study. However, rigorous approaches such as those adopted by Molina et al., including a series of negative controls and stringent in silico removal of potential contaminants, have enabled the identification of bacterial genera specific to the testicular environment, composed by *Blautia*, *Cellulosibacter*, *Robinsoniella*, and *Wandonia* [40]. Prostate microbiota has also been investigated in patients with benign prostatic hyperplasia, as shown in Figure 1. One study reported the isolation of viable bacteria from prostatic tissue, with the phylum *Proteobacteria* predominating in collected samples [41]. A second study, utilizing fluorescence in situ hybridization adapted for low-biomass microbiota, demonstrated the dominance of *Firmicutes*, and confirmed the presence of a distinct local microbiota in the prostate compared to the urinary milieu [42]. Additionally, the penile skin microbiota has been examined. According to a systematic literature review, similar colonization patterns dominated by *Corynebacterium* and *Staphylococcus* spp., typical skin commensals, were consistently reported across studies [43]. As discussed in Figure 1, it has been proposed that the penile microbiota, particularly anaerobic bacteria, may stimulate the immune system. Thus, genera such as *Peptostreptococcus*, *Prevotella*, and *Dialister* have been associated with increased cytokine production, leading to the recruitment of HIV-susceptible CD4+ T cells to the inner foreskin and a higher risk of HIV infection [44]. In conclusion, studies characterizing the male genital tract microbiota reveal substantial variability in bacterial abundance. In low-biomass microbiota, the risk of contamination during sampling, DNA extraction, and amplification processes for next-generation sequencing is significantly increased [45]. To overcome this limitation, the inclusion of negative controls at every stage (e.g., sampling, sample processing, and PCR amplification) is essential to detect potential contaminants [38]. Finally, the divergent microbial compositions reported across studies may be attributed to several factors: the dynamic nature of the microbiota, geographical variation, differences in sample collection and DNA extraction methods, contamination, and varying data analysis approaches. Therefore, well-designed longitudinal studies across diverse ethnic groups, using standardized methodologies for sample processing and data analysis, are needed to reduce variability and ensure comparability between studies.

## 3. Environmental Factors Effects on Male Microbiota and Male Reproductive Health

The scientific community has shown growing interest in the effects of diet, lifestyle habits, and environmental exposure to several factors on general health, particularly male reproductive health. Since environmental factors’ effects on male microbiota are not the principal aim of this review, in the following paragraph, the authors report the main evidence available on this argument, focusing on endocrine-disrupting chemicals, pesticides, and plastics. All the reported evidence is summarized in Table 1.

### 3.1. Diet and Lifestyle

Excessive levels of ROS can damage both mitochondrial and nuclear DNA in sperm, leading to an increased risk of congenital malformations in offspring [46]. Smoking, alcohol consumption, psychological stress, obesity, and diets high in fats, sugars, and processed foods have been related to elevated oxidative stress and altered gut microbiota composition [47,48]. Therefore, investigating the interactions among lifestyle factors and male microbiota could help elucidate the mechanisms underlying paternal programming effects on offspring health. In a retrospective study of 770 men seeking fertility treatment, *Ureaplasma* spp. and *Gardnerella vaginalis* in urine and *Enterococcus* spp. in semen were associated with elevated ROS levels and increased sperm DNA fragmentation [49]. In male mice, tobacco exposure induced heritable DNA mutations in spermatogonial stem cells, reduced testicular weight, and was associated with higher abundances of *Firmicutes* and *Actinobacteria* in gut microbiota, as shown in Table 1 [50,51]. Oral supplementation with fermented black barley in mice could reduce the cigarette smoke-induced damage, significantly increasing total sperm motility, relative abundances of *Oscillospira* and *Ruminococcus,* and, lastly, restoring fecal metabolites involved in the biosynthesis of steroid hormones (e.g., estrone, dehydroepiandrosterone sulfate (DHEA sulfate), and progesterone) [51]. Current evidence of dietary habits’ effect on paternal microbiota has been mainly collected from animal models. As summarized in Table 1, studies on fecal microbiota transplantation (FMT) from obese mice showed the following in healthy recipients: (1) increased weight gain, (2) localized epididymal inflammation, (3) higher abundance of intestinal *Prevotella copri*, negatively correlated with sperm motility, and (4) reduced levels of *Ruminococcaceae_NK4A214_group*, correlated with altered bile acids and vitamin A metabolism and thinning of the seminiferous epithelium [52,53]. Similarly, other research investigating the effects of high-fat and high-sugar diets found consistent alterations in gut microbiota composition, regardless of host genotype [54]. Mice fed a low-protein diet (LPD) showed global sperm DNA hypomethylation, related to decreased expression of genes regulating folate metabolism in the testes [55]. Additionally, females mated with LPD males displayed impaired uterine immune responses, altered cell signaling, and defective vascular remodeling during pre-implantation [55]. For instance, a systematic review of twenty-nine articles (fifteen rat models, seven pig models, six mouse models, and one in vitro study of human distal colon cells) has shown that dietary protein influences gut microbiota composition by promoting protein fermentation and nutrient absorption, a process that may also affect seminal microbiota [56]. In boars, dietary fiber supplementation increased the abundance of beneficial microbes, while reducing harmful genera like *Turicibacter*, *Romboutsia*, and *Clostridium_sensu_stricto_1* [57]. These changes promoted short-chain fatty acid (SCFA) production, particularly acetate and butyrate, which play a crucial role in sperm motility and in the regulation of meiotic-to-post-meiotic transitions during spermatogenesis [57]. Microbial metabolites could influence not only spermatogenesis, but also the sperm epigenome. Thus, it was recently found that they are involved in epigenetic modifications, including DNA and histone methylation and acetylation [58], and that butyrate can inhibit histone deacetylases (HDACs) [59]. According to emerging evidence on the gut–brain axis, chronic stress and mental health conditions, including anxiety, depression, and irritable bowel syndrome (IBS), may influence gut microbiota composition, which in turn could impact semen quality and microbial communities [60,61]. It is hypothesized that the seminal microbiota may act as a biosensor of paternal environmental stress, producing signaling molecules (such as serotonin, gamma-aminobutyric acid, dopamine, nitric oxide, carbon monoxide, hydrogen sulfide, and sulfur dioxide) that modulate the endocrine, immune, and nervous systems, as well as the testicular microenvironment [62]. In conclusion, dietary habits and lifestyle could contribute to the transmission of microbiota-mediated epigenetic changes. However, all the cited evidence is based on animal models. Furthermore, despite growing interest in bioactive food compounds and their potential role in paternal programming in humans, both clinical and in vivo studies in this area remain limited [63]. Most available human studies evaluate probiotic supplementation or dietary intervention effects within a single spermatogenic cycle, lasting about 74 days, to assess the improvement in semen quality. However, observational studies assessing paternal bioactive food compound intake prior to conception and correlating it with fetal development and neonatal outcomes may offer valuable insights into understanding the alterations in sperm cell epigenetics involved in paternal programming.

### 3.2. Endocrine Disruptors and Pesticides

Endocrine-disrupting chemicals (EDCs), which are predominantly synthetic compounds found in various materials, can interfere with the endocrine system [64]. Nowadays, most studies exploring the impact of EDCs on male reproduction and microbiota have been conducted in animal models (Table 1). In a rat model, daily exposure to di(2-ethylhexyl) phthalate (DEHP) caused increased oxidative stress, via activation of the Nuclear factor erythroid 2-related factor 2 (Nrf2) pathway, altered testicular structure, and reduced serum testosterone levels [65]. Similarly, the offspring of rats exposed to 500 mg/kg of di-n-butyl phthalate (DBP) exhibited intergenerational testicular damage and an increased relative abundance of Bacteroidetes, Prevotella, and *P. copri* [66]. Bisphenol AF (BPAF), an analog of bisphenol A (BPA), is commonly used in manufacturing industries. Both prenatal and postnatal BPAF exposure adversely affects the male reproductive system, according to several mechanisms: (1) decreased LH and follicle-stimulating hormone (FSH) levels, through upregulation of Kiss1 and binding to estrogen receptor α (ERα) in the hypothalamus, (2) altered expression of key steroidogenic enzymes such as StAR, CYP11A, 3β-HSD, and CYP17A1 in Leydig cells, (3) increased aromatase activity and estradiol production, (4) augmented ROS production, and (5) cytoskeletal disruption in Sertoli Cells, compromising the BTB [67]. Furthermore, maternal BPAF exposure during pregnancy could trigger innate and adaptive immune responses in the testes of adult offspring, leading to reproductive damage [67]. In male mice, BPA exposure led to increased abundance of intestinal Prevotellaceae, Akkermansia, and Methanobrevibacter, involved in mucosal integrity and butyrate production [68]. Another study demonstrated that BPA-induced gut dysbiosis, summarized in Table 1, was related to reduced levels of reproductive hormones (e.g., DHT, FSH, LH, estradiol, inhibin B, and testosterone), lower testicular fructose, disrupted testicular architecture, decreased sperm count, and sperm abnormalities [69]. As reported in Table 1, chlorinated polyfluorinated ether sulfonic acid (Cl-PFESA) exposure significantly altered hormonal serum levels and the gene expression involved in testicular steroidogenesis in a rat model [70]. Furthermore, Cl-PFESA exposure disrupted 47 gut metabolites, including short-chain fatty acids, bile acids, and amino acids [70]. Among pesticides, chlorpyrifos (CPF) exposure altered gut hormone release (e.g., pancreatic polypeptide, ghrelin, peptide YY, and glucagon-like peptide-1) and induced systemic inflammation [71]. Moreover, CPF exposure led to an increased abundance of *Enterococcus* spp., *Clostridium* spp., *Staphylococcus* spp., and *Bacteroides* spp., alongside decreased levels of *Lactobacillus* spp. and *Bifidobacterium* spp., according to several animal studies [72]. Glyphosate (GLY), a controversial herbicide, is commonly found in soil and water. Exposure to GLY reduced populations of beneficial bacteria such as Lactobacillus and decreased testosterone levels in avian species [73], while it significantly altered the relative abundance of Bacteroidetes and Firmicutes in rats (Table 1) [74]. Increased levels of Prevotella_1 and Bacteroides were negatively correlated with sperm quality, potentially via IL-17α pathway activation. This inflammatory response, mediated by T helper 17 A cells (Th17A) derived from the gut microbiota, may lead to testicular injury [73,74]. In conclusion, all these findings support the possible effect of EDCs in promoting systemic and testicular inflammation, through the induction of gut microbiota dysbiosis. Understanding the effects of such exposure in humans represents a highly challenging area of research, given that the severity and nature of male reproductive damage may vary depending on the species, timing, dose, and duration of exposure [67]. For this reason, the clinical relevance of this body of literature still needs to be investigated through observational human studies with controlled and standardized exposure.

### 3.3. Plastics

Microplastics (MPs), an emerging environmental contaminant, are increasingly associated with testicular disorders in mammals (Table 1). According to a study in mice, after FMT involving polystyrene MPs, the abundance of *Bacteroides* and *Prevotellaceae_UCG-001* increased, promoting testicular damage and the translocation of Th17 cells [75]. Polyethylene nanoplastics (NPs) have been shown to significantly impair male reproductive function across generations by altering the expression of key miRNAs [76]. In addition, these NPs caused a bacterial shift in gut microbiota composition, with decreased levels of *Allobaculum* and increased levels of *Desulfovibrio (C21_c20)* and *Ruminococcus gnavus*, which are positively associated with spermatogenesis [77]. Lastly, according to a murine study, long-term NP exposure may impair spermatogenesis, through the enrichment of microbial metabolites related to lipid metabolism in both testicular and intestinal metabolomes [78].

**Table 1 biology-14-01002-t001:** Summary of most relevant evidence on the impact of external factors on microbiota composition and male reproductive health.

Environmental Factors	Specific Exposure	Study Type/Population	Microbiota Changes	Clinical Impact on Male Reproduction Health	References
Diet	HFD	Mice model	>15% abundance of intestinal *Prevotella copri* related to reduced sperm motility	Altered spermatogenesis	Ding et al. [52]
	LPD	Mice model	Higher abundance of fecal *Enterococcus* and *Lactobacillus* and significantly reduced levels of fecal *Bifidobacterium* in the male offspring of mice fed LDP	Global sperm DNA hypomethylation, decreased gene expression for folate metabolism	Watkins et al. [55]
	Fiber supplementation	Boar model	Increased abundance of *Lactobacillus*, *Ruminococcus*, *Rikenellaceae_RC9_gut_group*, and *UCG-005*, reduced abundance of *Turicibacter*, *Romboutsia*, and *Clostridium_sensu_* *stricto_1*	Higher levels of acetate and butyrate, involved in spermatogenesis and sperm motility	Lin et al. [57]
Smoke	Cigarette-smoke	Mice model	Significantly increased relative abundances of *Firmicutes* and Actinobacteria and decreased relative abundances of *Bacteroidetes* and *Proteobacteria* in gut microbiota	Reduced size and weight of testes, reduced sperm motility	Zhong et al. [51]
EDCs	DEHP	Rats model	Major differences in the microbial profile in jejunal tract in DEHP group (*Lactobacillus*, *Streptococcus*, *Gemella*, *Mycoplasmataceae*, and *Rothia*—the most prevalent genera)	Induced testicular damage and significantly reduced serum levels of testosterone and LH	Zhao et al. [65]
	DBP	Rats model	Increased relative abundance of *Bacteroidetes*, *Prevotella*, and *P. copri*	Intergenerational testicular damage, such as increased seminiferous atrophy and spermatogenic cells apoptosis	Zhang et al. [66]
	BPA	Rats model	*Alloprevotella*, *uncultured_organism*, and *Prevotellaceae_UCG_* *001*—dominant genera in medium-dose BPA-exposed groups; *Parasutterella* was significantly higher in high-dose BPA-exposed groups	Reduced levels of DHT, FSH, LH, estradiol, inhibin B, and testosterone, lower testicular fructose, disrupted testicular architecture, decreased sperm count, and sperm abnormalities	Liu et al. [69]
	Cl-PFESA	Rats model	Significantly higher levels of *Ruminococcaceae* and *Desulfovibrionaceae* and significanty lower levels of *Pasteurellaceae* and *Micrococcaceae* after 6:2 Cl-PFESA exposure	Significantly altered serum levels of testosterone, progesterone, and cortisol	Zhao et al. [70]
Pesticides	CPF	Rats model	More evident dysbiosis in rats fed HFD and exposed beginning at newly weaned; specifically, CPF depletion of the relative abundances of *unclassified_f Ruminococcaceae*, *Oscillibacter*, *Paenalcaligenes*, and *Peptococcus* and enrichment of *Escherichia-Shigella*	Decreased serum concentrations of LH, FSH, and testosterone, and induced systemic inflammation, (elevated monocyte chemoattractant protein-1, TNF-α, and IL-6)	Li et al. [71]
	GLY	Avian model	Decreased abundance of *Firmicutes* and *Lactobacillus*; increased abundance of *Actinobacteria*	Decreased testosterone levels both at puberty and after 52 weeks of exposure	Ruuskanen et al. [73]
		Rats model	Significantly increased relative abundance of *Bacteroides* and *Prevotella_1*; significant decrease in the relative abundance of *Firmicutes*	Reduction in spermatogenic cells in seminiferous tubules, malformed nuclei in Sertoli cells, decreased sperm motility, and affected spermatogenesis	Liu et al. [74]
Plastics	Polystyrene MPs	Mice model (FMT)	Increased abundance of *Bacteroides* and *Prevotellaceae_UCG-001*	Altered testicular structure and destruction of BTB, significantly decreased levels of LH, FSH, and testosterone, and higher sperm abnormality	Wen et al. [75]
	Polystyrene NPs and amino-modified polystyrene NPs	Mice model	Higher abundance of *Desulfovibrio* and *Lachnospiraceae_NK4A136_group* and significantly decreased levels of *Blautia* and *Parabacteroides*	Impaired spermatogenesis	Zhou et al. [77]
	Polyethylene NPs	Mice model	Increased abundance of *Desulfovibrio (C21_c20)* and *Ruminococcus (gnavus)* and decreased abundance of *Allobaculum*	Intergenerational induced histological damage in the testicular tissue and altered sperm quality, sex hormone synthesis, and spermatogenesis	Sun et al. [76]

Acronym: HFD: high-fat-diet; LPD: Low-protein diet; EDCs: endocrine-disrupting chemicals; DEHP: di(2-ethylhexyl) phthalate; DHT: dihydrotestosterone; FSH: follicle-stimulating hormone; LH: luteinizing hormone; DBP: di-n-butyl phthalate; BPA: bisphenol A; Cl-PFESA: Chlorinated polyfluorinated ether sulfonic acid; CPF: chlorpyrifos; TNF: tumor necrosis factor; IL: interleukin; GLY: glyphosate; MPs: microplastics; NPs: nanoplastics; FMT: fecal microbiota transplantation; BTB: blood testes barrier.

## 4. Paternal Microbiota Effect on Offspring Health

Maternal microbiota has been considered the main determinant of early-life gut microbiome; however, growing evidence is highlighting the critical role of paternal microbial transmission. Maternal seeding during birth and breastfeeding has long been considered the primary source of microbial colonization [78], however, recent metagenomic studies have identified fathers as equivalent players in the development of the infant gut microbiota, especially in the context of cesarean deliveries. To assess the respective roles of both parents, Dubois et al. analyzed two longitudinal metagenomic datasets, the Health and Early Life Microbiota (HELMi) study and the SECFLOR study, both including cesarean-born infants who received maternal FMT [79]. The authors found that the maternal microbial contribution peaks during birth and declines over time, particularly following cesarean delivery [79]. Meanwhile, the paternal contribution stayed stable independently of the delivery mode, and cumulative input of microbial strains inherited from fathers was similar to the maternal contribution by the end of the first year of life [79]. Lastly, the microbial strains inherited by both parents minimally overlapped, suggesting that each parent provides distinct microbial lineages that synergistically shape the infant gut microbiome [79]. Previous results were obtained from the meta-analysis of over 1900 available metagenomic shotgun samples obtained from neonates, infants, adolescents, and their parents to calculate the cumulative relative abundance of shared microbial strain in the infant gut [80]. During the first two years of life, maternal strains accounted for a greater proportion of the infant microbiota (24.89 ± 30.87% relative abundance; 1.81 ± 1.63 strains) compared to paternal strains (0.52 ± 2.32% relative abundance; 0.29 ± 0.72 strains). These maternal contributions remained relatively stable during later infancy, adolescence, and adulthood [80]. Paternal strains increased later in life, with relative abundances of 7.25 ± 8.4% and 3.97 ± 3.61 in older children and adults, indicating long-term paternal microbial influence that extends beyond early infancy [80]. These results were supported by other studies that confirmed father-to-child microbial transmission in children aged 2–10 years, and higher father–child strain sharing compared to that between mother and children above 3 years of age [81,82]. In conclusion, maternal transmission seems to be prevalent during the early phase of microbiota development, while the paternal microbiome plays an increasingly relevant role over time. In the following sections, we report the evidence of paternal shaping of the infant microbiota and the consequent effects as regards behavioral disorders, metabolic health, and long-term health outcomes.

### 4.1. Offspring Neurodevelopment and Behavior

The impact of maternal obesity on the cognitive function and mental health of offspring has been extensively investigated, with studies reporting neurobiological alterations in neuronal morphology within the hippocampus and amygdala [83]. More recently, the effect of paternal diet on offspring’s higher-order cognitive functions has also been explored to elucidate underlying neurobiological mechanisms. Evidence from rodent models suggests that paternal caloric restriction and aerobic exercise prior to conception are associated with improved cognitive performance and reduced anxiety-like behaviors in the progeny [84]. Conversely, a paternal HFD has been linked to disrupted glucocorticoid signaling and increased anxiety-related behaviors in adult offspring [85]. Dietary-induced alterations in the gut microbiota may affect paternal epigenetic inheritance of cognitive functions, playing a role in neurodevelopment and behavioral outcomes in offspring.

In rodent models, increased anxiety-like behavior has been reported in offspring sired by males fed a HFD [86]. In contrast, a recent community-based study identified an inverse association between paternal obesity and major depressive disorder (MDD) in offspring (odds ratio [OR] = 0.70, 95% confidence interval [CI] = 0.54–0.91) [87]. However, a parental history of MDD was positively associated with increased odds of offspring obesity (OR = 1.74, 95% CI = 1.24–2.46), indicating complex and potentially bidirectional intergenerational links between mood disorders and metabolic traits [87]. Experimental data from Bodden et al. demonstrated that male mice exposed to a Western-style high-fat, high-sugar diet during adolescence sired offspring with increased birth weights and increased gut *Actinobacteria* abundance [88]. No differences were observed in anxiety-like behavior, locomotion, spatial or object learning, social interaction, or risk assessment, but offspring exhibited reduced behavioral despair during test [88]. In another murine study, males treated with antibiotics for four weeks exhibited significant depletion of gut bacteria, including *Clostridia_UCG_014_61* and several *Muribaculum* species [89]. Although the offspring’s gut microbiome remained unchanged, female offspring displayed increased anxiety- and depression-like behaviors, potentially mediated by altered expression of eight distinct PIWI-interacting RNAs (piRNAs) [89]. These piRNAs are hypothesized to regulate downstream gene networks, thereby influencing epigenetic inheritance mechanisms [89]. Complementary findings in rats revealed that paternal HFD exposure was negatively correlated with maternal investment and offspring weight throughout life [90]. Paternal HFD exposure induced increased anxiety-like behavior in adult offspring and marked changes in gut microbial alpha diversity, including an elevated *Firmicutes-to-Bacteroidetes* ratio and increased levels of *Bifidobacterium pseudolongum*, a beneficial bacterium, probably as a compensatory response within the gut microbiota to elevated levels of *Firmicutes* [90]. Moreover, recent work by Zhang et al. [91] showed that paternal methamphetamine exposure induced depressive-like behaviors in both first and second-generation male mice. These effects were associated with specific alterations in gut microbial composition, notably increased abundance of *Eubacterium_ruminantium_group*, *Enterorhabdus*, *Alloprevotella*, and *Parabacteroides* [91]. This evidence linking paternal microbiota to offspring neurodevelopment and behavioral outcomes is currently derived from animal studies. While these models offer valuable mechanistic insights, their translational relevance to humans remains limited due to differences in environmental exposures and developmental timelines. To advance our understanding in this field, well-designed human studies are needed, using standardized diagnostic criteria to evaluate neurodevelopmental alterations, along with sufficiently long follow-up periods. This is essential, as many neurodevelopmental and behavioral disorders manifest at specific developmental stages during childhood.

### 4.2. Offspring Metabolic Health

Recent findings have suggested that paternal dietary patterns and microbiota composition prior to conception could influence offspring metabolic outcomes. The shifts in microbial populations and their associated metabolites may have transgenerational effects through epigenetic mechanisms, modulating metabolic pathways. In rodent studies, offspring (F2 generation) of rats subjected to a high-fat, high-sucrose, and high-salt diet (HFSSD) over two generations (F0 and F1) exhibited liver function abnormalities and altered lipid profiles in adulthood [92]. These metabolic disruptions were accompanied by significant changes in the relative abundance of specific gut bacterial genera in the F2 offspring of both sexes [92]. In contrast, paternal adherence to a high-protein diet produced favorable metabolic effects in offspring. Rats fed a high-protein diet showed improved glucose tolerance, and this diet enhanced insulin tolerance in both male and female adult offspring [93]. Furthermore, the abundance of *Muribaculaceae*, a bacterial genus inversely associated with fat mass, was increased in the adult gut microbiota of both sexes [93]. Dietary supplementation prior to conception has also shown promise in improving offspring metabolic profiles. In a study involving paternal exposure to an oligofructose-supplemented diet for nine weeks, female adult offspring showed increased circulating levels of peptide tyrosine tyrosine (PYY), a gut hormone linked to satiety and reduced hepatic triglyceride accumulation [94]. Moreover, gut microbiota analysis revealed a higher abundance of Bacteroidetes in female offspring, a phylum known to produce SCFAs that support mucosal integrity and metabolic homeostasis. In male offspring, increased levels of *Christensenellaceae*, a family generally not associated with metabolic syndrome, were observed at nine weeks of age [94]. In another study, male mice were fed a HFD for 16 weeks, followed by a 12-week treatment with the probiotic *Lactobacillus rhamnosus*. The probiotic treatment induced notable changes in the sperm epigenetic signature, as well as alterations in miRNA expression, DNA methylation, and histone modifications in the liver and pancreas of treated mice compared to untreated controls [95]. Similar epigenetic changes were also observed in the offspring of probiotic-treated males, suggesting potential intergenerational effects [95]. For this reason, nutritional support and microbiome modulation prior to conception may represent a crucial opportunity for intergenerational disease prevention. However, as previously described, human data are still missing.

### 4.3. Offspring Growth and Gastrointestinal Disease

Initial work by Ding et al. [96] demonstrated that a history of in utero exposure to 2,3,7,8-tetrachlorodibenzo-p-dioxin (TCDD) resulted in impaired sperm quality in male mice. This was associated with an increased risk of preterm birth, intrauterine growth restriction, and higher susceptibility to necrotizing enterocolitis (NEC) in their offspring. The observed effects were attributed to placental dysfunction linked to paternal exposure [96]. Subsequently, the same group investigated whether dietary interventions could mitigate these intergenerational outcomes. In TCDD-exposed males, preconceptional supplementation with fish oil led to significant alterations in the gut microbiota of offspring. Notably, there was a reduction in the relative abundance of *Firmicutes*, *Veillonella*, and *Megasphaera*, alongside an increase in *Negativicoccus massiliensis* [97]. These microbial shifts are notable, given that a prospective study involving stool samples from 122 very-low-birthweight infants revealed a relative paucity of strict anaerobes, particularly members of the *Negativicutes* class, to be associated with the development of NEC [98]. Further expanding on the interplay between paternal gut microbiota and offspring development, Argaw-Denboba et al. introduced the novel concept of the “gut–germline axis” [10]. This study first demonstrated that perturbations of the gut microbiota, induced through ad libitum administration of non-absorbable antibiotics (nABX), elicited significant reproductive effects in male mice. These included the following: 1. smaller testes, lower sperm count, and increased sperm abnormalities; 2. dysregulated metagenomic pathways (e.g., sphingolipids, glycerophospholipids, and endocannabinoids), all known to be crucial for germ cell function; and 3. downregulation of leptin [10]. Offspring, sired by fathers with gut dysbiosis, exhibited consistent phenotypes such as lower birth weight, growth retardation, and increased postnatal mortality. These effects were present only during paternal microbiota dysbiosis and were reversible upon microbiome restoration. To investigate underlying mechanisms of paternal inheritance, the same group found that sperm is the primary medium for transmitting dysbiosis-induced phenotypes. Indeed, it was demonstrated that in vitro fertilization (IVF) progeny sired by dysbiotic sperm donors showed neonatal birth weight and impaired postnatal growth. Sperm-borne small RNAs were significantly altered in response to gut dysbiosis, particularly miR-141 and miR-200a, which co-regulate epithelial–mesenchymal transition and placental development. Moreover, tRNA-derived fragments (tRF-Gly-GCC), which have been implicated in intergenerational epigenetic inheritance, were upregulated. Finally, offspring exhibited placental dysfunction, including impaired vascularization, abnormal fetoplacental ratios, downregulation of placental growth factor (PLGF), and upregulation of biomarkers used in human preeclampsia screening [10]. This study represents an important starting point for future research on microbial inheritance, discovering the intricate epigenetic mechanisms involved in paternal programming.

## 5. Main Findings, Research Gaps, and Further Perspectives

Lifestyle and environmental factors may increase the risk of chronic diseases in subsequent generations through parental influences [99]. One proposed mechanism by which paternal exposure prior to conception affects offspring health is epigenetic inheritance [63]. The sperm epigenome comprises a pattern of epigenetic signatures, including sperm DNA methylation, which plays a role in cell differentiation and embryonic development, histone modifications, which protect sperm DNA from external stressors, and small non-coding RNAs (sncRNAs), which regulate gene expression [100]. In this context, commensal microbes, especially those residing in the gut and male reproductive tract, may modulate host epigenetics through non-covalent modifications, histone remodeling, DNA methylation changes, and regulation of immune cells via microbial-derived metabolites [101]. These microbe-induced epigenetic modifications in the seminal environment may mediate paternal programming effects on the offspring, thereby influencing developmental trajectories and long-term health outcomes. As highlighted in the reviewed literature, most of the explored mechanisms focus on microbial-induced epigenetic modifications in the father that affect the progeny. Additional proposed pathways, such as the action of bEVs or microbial signaling molecules, may also indirectly mediate paternal effects in response to microbial perturbation.

Nonetheless, there are notable limitations. As a narrative review, the included literature was selected based on relevance. This non-systematic approach may introduce selection bias and could overlook studies presenting conflicting evidence. Moreover, the molecular pathways through which the paternal microbiota modulates gamete epigenetics and offspring phenotypes remain largely unexplored. Firstly, most available data are derived from animal models, where microbial shifts were induced by FMT, antibiotic, or toxin administration. Then, in selected studies, the authors did not evaluate the possible contamination of seminal microbiota with maternal microbiota, potentially altering the in utero environment and the sperm epigenome. To bridge these gaps, longitudinal human studies are essential. These should assess paternal microbiome composition before conception and follow offspring development across multiple life stages to validate transgenerational effects. Interventions aimed at modulating the paternal microbiota, through diet, prebiotics, or probiotics, require standardized and replicable protocols through large randomized clinical trials. Dietary interventions should be tightly controlled and based on validated Food Frequency Questionnaires (FFQs) to study more homogeneous populations and account for major confounding factors such as caloric intake, nutrient supplements, and diet type. Then, the duration of probiotic or dietary interventions to induce beneficial epigenetic modifications in sperm needs to be investigated. Additionally, appropriate follow-up durations are needed to assess long-term impacts on child health. Finally, understanding how paternal and offspring microbiomes, including those in the skin, urinary tract, and oral cavity, interact within shared environments later in life may further elucidate the dynamics of paternal programming.

## 6. Conclusions

This review aimed to summarize current evidence on the molecular mechanisms through which the paternal microbiome may influence offspring development, with a focus on the emerging concept of paternal programming. Animal models, with tightly controlled environmental exposures and the use of histological and epigenomic mapping techniques, have begun to uncover the intricate interplay between paternal microbiota, sperm epigenetics, and offspring phenotypes. Using the same experimental models, future research should explore the role of paternal programming mediated by the seminal microbiota both before and after copulation, particularly in relation to its interaction with the vagino-uterine microbiome, forming the so-called “complementary semino-vaginal microbiota”. Moreover, investigations into sncRNAs, microbiota-derived metabolites, and their interactions with the maternal immune system during embryo implantation may help identify key mediators of paternal programming. To date, human studies have only examined microbial strains inherited from both parents in shaping the infant gut microbiota. To our knowledge, no studies have yet evaluated the paternal microbiota before conception in relation to specific offspring outcomes. Given the complex mechanisms linking environmental exposures to microbiome alterations, there is a critical need for large, standardized randomized clinical trials to assess the effectiveness of periconceptional interventions. By understanding paternal programming in humans, clinicians may ultimately develop targeted preconception care strategies, aimed at improving reproductive outcomes and reducing the burden of chronic disease in future generations.

## Figures and Tables

**Figure 1 biology-14-01002-f001:**
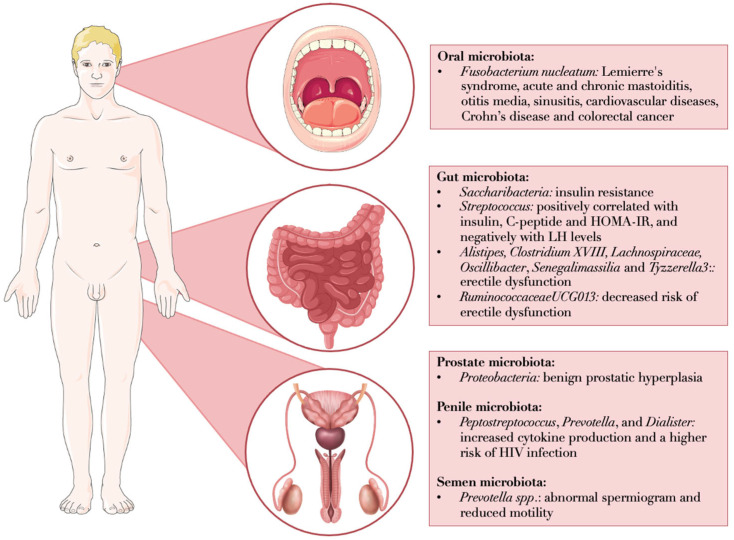
Microbial alterations across anatomical sites associated with male reproductive and general health conditions. *Fusobacterium nucleatum*, a prevalent Gram-negative anaerobe in human dental plaque, has been implicated in the activation of the host immune response and various systemic pathological conditions. In the gut, according to the GELDING theory, bacterial endotoxins may impair Leydig cell steroidogenesis and LH signaling from the pituitary, leading to reduced testosterone production. Several gut microbial species have been associated with metabolic and reproductive health, including links to sexual dysfunction. The male genital tract microbiome is challenging to characterize due to its low biomass and susceptibility to contamination. However, it has been indirectly investigated through studies on semen microbiota. Among these, *Prevotella* has been frequently associated with abnormal semen parameters and reduced sperm motility, although findings remain inconsistent and require further validation. This figure was originally created using Microsoft Office PowerPoint. Acronyms: LH: luteinizing hormone; HOMA-IR: homeostatic model assessment of insulin resistance.

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
