# Peer review of "The Father’s Microbiome: A Hidden Contributor to Fetal and Long-Term Child Health"

_biology, 2025, doi:10.3390/biology14081002_

Round 1
Reviewer 1 Report
Comments and Suggestions for Authors
Thank you for submitting the manuscript "Paternal microbiota contribution to fetal health" to Biology. I have conducted a thorough technical review of the content for clarity, scientific rigor, logical structure, and linguistic adequacy. The following are the main points and suggestions for improvement, organized by type of adjustment:
- In some passages, the article conveys an overly affirmative impression about the impact of the paternal microbiota, even when the evidence comes primarily from animal models.
- Phrases such as "the paternal microbiota influences offspring neurodevelopment" do not consider the strength of the evidence.
- The topic of the "gut–germline axis" is mentioned several times with similar definitions.
- Some sections start abruptly, such as section 3.1 ("Diet and lifestyle") and section 4.1 ("Offspring neurodevelopment and behavior").
- There are minor grammatical errors and inconsistencies in the use of articles, prepositions, and verb tenses (e.g., "the paternal contribution stay stable" → "stays stable").
- Some acronyms appear before they are defined (e.g., "EDCs", "BTB", "SCFA").
- The conclusion could better reinforce the methodological limitations of the cited studies.
- Figure 1 is mentioned, but the caption could be more informative, briefly explaining what the reader should look for regarding microbial changes and their reproductive impact.
Author Response
Thank you for submitting the manuscript "Paternal microbiota contribution to fetal health" to Biology. I have conducted a thorough technical review of the content for clarity, scientific rigor, logical structure, and linguistic adequacy. The following are the main points and suggestions for improvement, organized by type of adjustment:
- In some passages, the article conveys an overly affirmative impression about the impact of the paternal microbiota, even when the evidence comes primarily from animal models.
Thank you for your suggestions. We carefully revised the manuscript to ensure strength and limitations of the available evidence. We explicitly stated when findings derive from animal studies, to prevent any overgeneralization to humans and to highlight the need for further research in “Research Gaps and Further Perspectives” section.
- Phrases such as "the paternal microbiota influences offspring neurodevelopment" do not consider the strength of the evidence.
In the revised manuscript, we carefully reviewed and modified the phrasing throughout the text to more accurately reflect the current state of evidence, which is primarily based on animal models. We replaced overly definitive statements with more cautious language, using modal verbs such as “may,” “could,” or “might” to acknowledge the preliminary nature of these findings.
- The topic of the "gut–germline axis" is mentioned several times with similar definitions.
In the revised version of the manuscript, we refer to the gut–germline axis in the “Offspring Growth and Gastrointestinal Disease” section to highlight the relationship between paternal intestinal microbiota alterations and fetal outcomes (line 568-577). We then revisit the concept in the “Research Gaps and Further Perspectives” section, where we explore in greater depth the mechanisms underlying transgenerational inheritance as investigated by the same research group (line 609-619). The references to the gut–germline axis serve different purposes in each section and are intended to provide both context and mechanistic insight.
- Some sections start abruptly, such as section 3.1 ("Diet and lifestyle") and section 4.1 ("Offspring neurodevelopment and behavior").
We included an opening paragraph in both sections. In the “Diet and Lifestyle” section (lines 300–308), we introduce the topic by presenting established evidence on ROS production related to external exposures and their subsequent effects on sperm function and offspring health. In the “Offspring Neurodevelopment and Behavior” section (lines 484–494), we begin by discussing the known effects of maternal and paternal diet on neurological development, and subsequently examine how these effects may be modulated by microbiota.
- There are minor grammatical errors and inconsistencies in the use of articles, prepositions, and verb tenses (e.g., "the paternal contribution stay stable" → "stays stable").
We extensively checked the grammar of text, table and figure.
- Some acronyms appear before they are defined (e.g., "EDCs", "BTB", "SCFA").
We carefully checked all cited acronyms and ensured they are defined upon their first appearance in the text.
- The conclusion could better reinforce the methodological limitations of the cited studies.
We addressed the limitations of both this review and the cited studies in detail in the section entitled “Research Gaps and Further Perspectives” (lines 629- 642). Lines 634–638 clearly describe the methodological limitations of the available studies. In the Conclusion section, we emphasized that “the interplay between paternal microbiota, sperm epigenetics, and offspring phenotypes is beginning to be investigated,” and we discussed directions for future research needed to apply these findings into clinical practice.
- Figure 1 is mentioned, but the caption could be more informative, briefly explaining what the reader should look for regarding microbial changes and their reproductive impact.
As requested, we have revised the caption of Figure 1 to provide a more informative description (line 90-104). The updated caption briefly highlights key microbial species and outlines their proposed impact on host reproductive health, including mechanisms involving immune activation, hormonal signaling, and sperm quality.
Reviewer 2 Report
Comments and Suggestions for Authors
Title
Please, rewrite the title of the manuscript. Maybe this one is more suitable? The Father’s Microbiome: A Hidden Contributor to Fetal and Long-Term Child Health? Discuss with colleagues.
Simple summary
Please, rewrite a simple summary. It’s difficult to read and understand right now. You begin with discussing man microbiota, and then unexpectedly discuss child microbiome and then, again back to mechanisms. Use this plan of ss: Introductory sentence (Fathers impact on offspring health- why is it important? ). Mention factors involved in it: toxins, diet, microbiome, sperm and hormones. Write a couple of sentences about epigenetics and gut-germline signalling connection? In conclusion, write about problems and perspectives, and lack of human data.
Key words
Think of changing key words: Male microbiome, Seminal microbiome, Sperm epigenetics, Paternal programming, Offspring health, Paternal epigenetic inheritance, Preconception health, Intergenerational metabolic programming, Microbial dysbiosis
Abstract
Rewrite the abstract. The first sentence is too broad. It repeats the fact of maternal dominance and gives no information about understudied paternal roles. Rewrite it so that it highlights the novelty of paternal microbiota research.
What do you mean “male microbiota”? Is it gut or semen? Please, clarify it.
Biological mechanisms mentioned in the abstract are too broad, please , specify them ( like sperm miRNA alterations?).
In conclusion you write about lack of human data, but provide no mention of present animal data. The aim overlaps with the rest of the abstract. Please, write what new insights the review offers for readers. Use short sentences. Now the abstract contains long, dense and difficult to read sentences.
Introduction
The first sentences use too general terms. It unfocuses the reader from the idea of paternal-fetal investigation. The shift from maternal to paternal microbiome is abrupt. There is a contradiction between the line “ Male microbiota has been largely investigated in the reproductive field” with previous emphasis on maternal dominance. So why to study paternal-fetal health?
In text you jump between oreal, gut and semen microbiota with no discussion of their interactions. You don't obviously state what is unknown in this field or why it is important scientifically or clinically. Reader misses the core idea of how they interact. The key points are just listed without connecting logic.
The aim overlaps the abstract and does not explain why this review advances the research field. The intro concludes with academic role and provides no perspectives for clinical application. How could this research and analysis improve prenatal health care?
Male microbiota
Gut microbiota
The text is too dense, textbook like description , without mentioning why this matters without explaining why this matters for paternal-fetal health. You jump abruptly from androgen-microbiota interactions to leaky gut hypothesis next to insulin resistance and then to erectile dysfunction without clear transition. rewrite it. THe chaptec cites many studies, but does not explain their interconnection to impact on offspring health.
Among bacterial groups, please, clarify what groups are the most important and impactful. Provide a summary paragraph. Add conclusions with clinical implications.
Discussion
The first sentence is too broad and does not provide the novelty of your review findings. You mention "human studies indicate..” What studies? How robust is the evidence?
Your emphasis on gut-germline axis without discussion how sperm transmits signals and what epigenetic factors are implicated. Discuss intervention potential. Discuss evolutionary context: how parental-offspring microbiome inheritance might evolve?
Prioritize the list of future research needs,
Add concluding statement.
Conclusion
Much of the information was already mentioned in the discussion section. The first sentence repeats the common knowledge, without emphasizing your review importance and role of paternal microbiome studies.
Lines 577-588 mention methodological weaknesses without discussing progress and solutions. The terms complex pathways and molecular mediators are too broad.The conclusion does not highlight the missing mechanisms.
Conclusion ignores broader implications of microbiome research. The text lacks conclusive sentences that clarifies the review importance and motivates future research.
Author Response
Title
Please, rewrite the title of the manuscript. Maybe this one is more suitable? The Father’s Microbiome: A Hidden Contributor to Fetal and Long-Term Child Health? Discuss with colleagues.
Dear Reviewer, thank you for your suggestion. My colleagues and I discussed the possibility of changing the title as you recommended, and we all agreed to adopt your proposal.
Simple summary
Please, rewrite a simple summary. It’s difficult to read and understand right now. You begin with discussing man microbiota, and then unexpectedly discuss child microbiome and then, again back to mechanisms. Use this plan of ss: Introductory sentence (Fathers impact on offspring health- why is it important? ). Mention factors involved in it: toxins, diet, microbiome, sperm and hormones. Write a couple of sentences about epigenetics and gut-germline signalling connection? In conclusion, write about problems and perspectives, and lack of human data.
In the revised manuscript, we have largely revised the simple summary to improve clarity, according to the structure you recommended.
Key words
Think of changing key words: Male microbiome, Seminal microbiome, Sperm epigenetics, Paternal programming, Offspring health, Paternal epigenetic inheritance, Preconception health, Intergenerational metabolic programming, Microbial dysbiosis
We changed the key words as you suggested.
Abstract
Rewrite the abstract. The first sentence is too broad. It repeats the fact of maternal dominance and gives no information about understudied paternal roles. Rewrite it so that it highlights the novelty of paternal microbiota research.
What do you mean “male microbiota”? Is it gut or semen? Please, clarify it.
Biological mechanisms mentioned in the abstract are too broad, please , specify them ( like sperm miRNA alterations?).
In conclusion you write about lack of human data, but provide no mention of present animal data. The aim overlaps with the rest of the abstract. Please, write what new insights the review offers for readers. Use short sentences. Now the abstract contains long, dense and difficult to read sentences.
We extensively changed the abstract. We clarified that current evidence of paternal programming primarily stems from animal model studies on intestinal microbiota alterations. We also specified key biological mechanisms involved, including sperm epigenetic modifications and hormonal signaling. Lastly, we emphasized the novelty and research gaps in this field, particularly the lack of longitudinal human studies.
Introduction
The first sentences use too general terms. It unfocuses the reader from the idea of paternal-fetal investigation. The shift from maternal to paternal microbiome is abrupt. There is a contradiction between the line “ Male microbiota has been largely investigated in the reproductive field” with previous emphasis on maternal dominance. So why to study paternal-fetal health?In text you jump between oreal, gut and semen microbiota with no discussion of their interactions. You don't obviously state what is unknown in this field or why it is important scientifically or clinically. Reader misses the core idea of how they interact. The key points are just listed without connecting logic.
The aim overlaps the abstract and does not explain why this review advances the research field. The intro concludes with academic role and provides no perspectives for clinical application. How could this research and analysis improve prenatal health care?
Thank you for these suggestions. We clarified the rationale for investigating paternal microbiomes. First, we provided an overview of the current knowledge on the role of male microbiota in reproductive health, distinguishing between the gut, oral, and seminal microbiota, and highlighting their potential interactions. Then we emphasized the possible role of paternal microbiota in pregnancy outcome and fetal health, with a smoother transition from the traditionally maternal-centric perspective. Furthermore, we introduced the possible mechanisms of paternal programming and the scientific relevance of this field, as suggested. We hope the revised Introduction now better reflects the novelty and significance of the review.
Male microbiota
Gut microbiota
The text is too dense, textbook like description , without mentioning why this matters without explaining why this matters for paternal-fetal health. You jump abruptly from androgen-microbiota interactions to leaky gut hypothesis next to insulin resistance and then to erectile dysfunction without clear transition. rewrite it. THe chaptec cites many studies, but does not explain their interconnection to impact on offspring health.
Among bacterial groups, please, clarify what groups are the most important and impactful. Provide a summary paragraph. Add conclusions with clinical implications.
We have thoroughly revised this part of the manuscript, as you suggested. We have improved the structure and flow of the paragraph, clarified key mechanisms linking the gut microbiota to male reproductive function. As cited at the beginning of paragraph “Male microbiota” (line 86-.87) the principal aim is to discuss the role of different sites microbiota in male reproductive health. Furthermore, we added a summary paragraph at the end of the section to highlight the relevance of the gut–testis axis and its clinical implications, as suggested.
We followed all these suggestions also to revise the paragraph “oral microbiota in male”, showing data on biunivocal effect of microbiota and hormonal balance and the evidence on the link among oral health status and mela reproductive health.
Discussion
The first sentence is too broad and does not provide the novelty of your review findings. You mention "human studies indicate..” What studies? How robust is the evidence?
Your emphasis on gut-germline axis without discussion how sperm transmits signals and what epigenetic factors are implicated. Discuss intervention potential. Discuss evolutionary context: how parental-offspring microbiome inheritance might evolve?
Prioritize the list of future research needs,
Add concluding statement.
We thank the reviewer for these valuable suggestions. In response, we have retitled the section as “Research Gaps and Further Perspectives” to better reflect its purpose and content. We revised the opening paragraph, starting from the evidence on the seminal epigenome and epigenetic signalling involved in paternal programming. Thern, we expanded the results of the study by Argaw-Denboba to better explain the gut–germline axis hypothesis, including mechanistic insights (small non-coding RNAs, tRNA fragments). Then, we clearly reported all the limitations of current evidence, underlying that the impact of gut microbial perturbations on sperm function and embryo development is referred only to animal studies.
In conclusion, we have clearly outlined and structured the major future research directions, including:
- Expanding microbiome research to less-studied paternal anatomical sites,
- Investigating paternal microbiota interactions with maternal reproductive microbiomes,
- Exploring epigenetic transmission pathways, and the other possible mechanisms involved
- Need for longitudinal human studies.
Lastly, we added a conclusive paragraph to this section, suggesting future clinical research on preconception modulation of the paternal microbiome through diet or probiotics.
Conclusion
Much of the information was already mentioned in the discussion section. The first sentence repeats the common knowledge, without emphasizing your review importance and role of paternal microbiome studies.
Lines 577-588 mention methodological weaknesses without discussing progress and solutions. The terms complex pathways and molecular mediators are too broad.The conclusion does not highlight the missing mechanisms.
Conclusion ignores broader implications of microbiome research. The text lacks conclusive sentences that clarifies the review importance and motivates future research.
We have significantly changed the Conclusion section. The revised conclusion now explicitly highlights the potential mechanisms linking the paternal microbiome and offspring development, including emerging evidence from animal models on sperm-borne epigenetic signals and their modulation by gut and seminal microbiota. We underlined the need to investigate other paternal anatomical sites (e.g., oral, skin, urinary tract), the role of the semino-vaginal microbiota and the potential interactions between microbiota-derived metabolites and the maternal immune system during implantation. Finally, we reinforced the clinical relevance of the topic by advocating for future studies on preconception care strategies and the potential to reduce chronic disease burden in future generations
Reviewer 3 Report
Comments and Suggestions for Authors
This review manuscript summarizes the current evidence on the influence of the paternal microbiota on male reproductive health and subsequent fetal and offspring outcomes and proposes a "gut-germline axis" as a key mechanism. This is a well-written, timely and comprehensive review of a rapidly growing field. The manuscript is well organized, the arguments are logical, and the conclusions are appropriately restrained. Below is some comment\questions \suggestions for the authors to consider.
- As a narrative review, the approach is to build a cohesive story from the available literature. This is appropriate for summarizing a developing field. The main limitation of a narrative review, which the author should acknowledge, is the potential for selection bias. Unlike a systematic review, there is no explicit, reproducible methodology for literature searching and inclusion. This can lead to the unintentional omission of conflicting studies. A brief sentence in the introduction or limitations section acknowledging this would strengthen the manuscript's transparency.
- The authors convincingly present the "gut-germline axis" as the central hypothesis. The proposed mediators include microbial metabolites (e.g. SCFAs), systemic inflammation (e.g. LPS) and epigenetic carriers (e.g. small non-coding RNAs in sperm). Based on the authors' comprehensive review, is a consensus emerging as to which of these pathways is the primary driver? Or is it more likely that different paternal exposures (e.g. diet or toxins) use different pathways to program the germline?
- Also, the authors may need to explain the mechanisms of the "gut-germline axis" in more detail. The proposed mechanisms (metabolites, inflammation, sncRNAs) are mentioned, but the level of evidence for each is not explicitly compared. The discussion could be deepened by examining in more detail which of these pathways is best documented.
- The review addresses the influence of the seminal microbiome on pregnancy outcomes (lines 201-207) and its possible interaction with the vaginal and endometrial microbiome. Could you speculate further on the role of the acellular seminal plasma itself? Does the microbial composition of semen directly alter the composition of signaling molecules, cytokines and extracellular vesicles in the seminal plasma, which then acts as a primary messenger to the maternal reproductive tract?
- The authors suggest that paternal health can be improved via interventions like diet or probiotics. From the literature, is there any indication of the timescale required for such interventions to be effective? Given that a full cycle of spermatogenesis takes approximately 74 days, would an intervention need to be sustained for at least three months prior to conception to have a meaningful impact on sperm quality and epigenetic programming?
- The review states that the contribution of the paternal microbiome becomes stronger as the child grows older, while the maternal contribution is strongest at birth. What could be the reason for this? Is this simply a result of increased physical contact and shared environment with the father over time, or could there be a more complex biological reason? For example, could it be that the paternal strains are better adapted to the more mature gut environment of an older infant, allowing them to displace the early maternal colonizers?
- At the end of the discussion, the authors briefly mention the "evolutionary significance" of paternal inheritance. I find that fascinating. Could the authors elaborate on what the evolutionary advantage of this microbial double inheritance by the parents might be? Does it provide the offspring with a more diverse microbial "toolkit" that prepares them for a broader range of environments, or does it serve as an honest signal of the father's fitness to the next generation?
- While the review expertly covers the gut, oral and semen microbiome, are there other, less studied paternal microbiomes (e.g. skin, urine) that could also play a significant, albeit smaller, role in shaping the infant's microbiome through direct contact after birth? Is there any preliminary evidence in this area?
- Here are minor errors and corrections
- – Line 53: Add full stop after “maternal domain [3,4]”
- – Line 56-57. The claim that "seminal microbiota could affect implantation and placental development" is not supported by the cited reference [7], which focuses on cytokine signaling in seminal plasma and its effect on sperm quality. There is no discussion of seminal microbiota or their influence on implantation or placental development in that source. I recommend either providing a more appropriate reference or rephrasing the statement to reflect the actual content of reference 7.
- - Line 95: "...Clostridium innocuum, express steroid-metabolizing enzymes..." The comma before "express" should be removed
- - Line 116: "...Tremellen et al [18]." - A period is missing
- - Line 156: "Campylobacter A" - The taxon "Campylobacter A" is unusual.
- - Line 272: "Ambiental exposure" - "Ambiental" should be "Environmental".
- - Page 6, Figure 1: In the "Gut microbiota" box, "insuline resistance" contains a typo. It should be "insulin resistance".
- Page 10, Table 1: "Biffidobacterium" should be corrected to "Bifidobacterium".
- Page 11, Table 1: "animo-modified polystyrene NPs" likely contains a typo and should be "amino-modified...".
- Page 11, Table 1: “Firmicutes” should be written in italic
- Page 12, Line 437: "the paternal contribution stay stable" should be corrected to "stays stable"
- – Ref #80 "programs ẞ 2-cell dysfunction" - Please check the original source (Ng et al., 2010). The correct term is likely "β-cell dysfunction". The "2" seems extraneous.
Author Response
This review manuscript summarizes the current evidence on the influence of the paternal microbiota on male reproductive health and subsequent fetal and offspring outcomes and proposes a "gut-germline axis" as a key mechanism. This is a well-written, timely and comprehensive review of a rapidly growing field. The manuscript is well organized, the arguments are logical, and the conclusions are appropriately restrained. Below is some comment\questions \suggestions for the authors to consider.
- As a narrative review, the approach is to build a cohesive story from the available literature. This is appropriate for summarizing a developing field. The main limitation of a narrative review, which the author should acknowledge, is the potential for selection bias. Unlike a systematic review, there is no explicit, reproducible methodology for literature searching and inclusion. This can lead to the unintentional omission of conflicting studies. A brief sentence in the introduction or limitations section acknowledging this would strengthen the manuscript's transparency.
Thanks you for your suggestions. We specified that this manuscript type is a narrative review in “Introduction” section (line 80) and in paragraph 5 we underlined the possible limitation of selecting articles, as you suggested (line 629-631).
- The authors convincingly present the "gut-germline axis" as the central hypothesis. The proposed mediators include microbial metabolites (e.g. SCFAs), systemic inflammation (e.g. LPS) and epigenetic carriers (e.g. small non-coding RNAs in sperm). Based on the authors' comprehensive review, is a consensus emerging as to which of these pathways is the primary driver? Or is it more likely that different paternal exposures (e.g. diet or toxins) use different pathways to program the germline?
Thank you for this relevant question. As discussed in the revised “Research Gaps and Further Perspectives” section (lines 622–628), most of the available studies (e.g., Argaw-Denboba 2024; Sheth 2022; Masson 2025) have investigated mechanisms centered on microbial-induced epigenetic modifications in the father, particularly following microbiota shifts induced by environmental exposures. Based on this evidence, we suggested that additional pathways, mediated by bEVs or microbial signaling molecules, may also contribute to paternal effects in response to microbial perturbations, but these remain insufficiently explored. Importantly, while each of the cited studies evaluated changes in the sperm epigenome following external exposure, they did not differentiate their findings based on the specific type of exposure. Furthermore, due to the limited number of available studies, no definitive conclusions can be drawn at this stage regarding which pathway is the primary driver. We also highlighted the potential role of confounding factors, such as the interaction with maternal microbiota, which may alter the intrauterine environment and consequently have an impact on offspring outcomes.
- Also, the authors may need to explain the mechanisms of the "gut-germline axis" in more detail. The proposed mechanisms (metabolites, inflammation, sncRNAs) are mentioned, but the level of evidence for each is not explicitly compared. The discussion could be deepened by examining in more detail which of these pathways is best documented.
The “gut-germline axis” has been proposed in the field of paternal programming by Argaw-Denboba and colleagues. According to your suggestions, we revised the concept in the “Research Gaps and Further Perspectives” section, where we explore in greater depth the mechanisms underlying transgenerational inheritance as investigated by the research group (line 609-619). - The review addresses the influence of the seminal microbiome on pregnancy outcomes (lines 201-207) and its possible interaction with the vaginal and endometrial microbiome. Could you speculate further on the role of the acellular seminal plasma itself? Does the microbial composition of semen directly alter the composition of signaling molecules, cytokines and extracellular vesicles in the seminal plasma, which then acts as a primary messenger to the maternal reproductive tract?
As you suggested, we included a review of the current knowledge on the role of acellular seminal plasma as an introduction to the topic addressed in the section “semen microbiota” (line 190-215). We then described the potential mechanisms proposed by available studies regarding the modulation by the seminal microbiome of signaling molecules, cytokines and extracellular vesicles in maternal reproductive tract (line 242-253).
- The authors suggest that paternal health can be improved via interventions like diet or probiotics. From the literature, is there any indication of the timescale required for such interventions to be effective? Given that a full cycle of spermatogenesis takes approximately 74 days, would an intervention need to be sustained for at least three months prior to conception to have a meaningful impact on sperm quality and epigenetic programming?
In the revised version of the manuscript, we have addressed the question regarding the timing and duration of paternal interventions, by summarizing evidence from recent literature suggesting that the effects on sperm quality are time-dependent (line 363-367). We have also expanded the discussion to include the current limitations in clinical and in vivo studies investigating the role of bioactive food compounds in paternal programming. Given the long timeframes required to establish cohorts of adult offspring, we emphasized the potential of alternative approaches such as observational studies on paternal dietary intake with a follow-up long enough to assess neonatal and child outcomes (line 639-642). These studies could help identify correlations between nutritional exposures, sperm epigenetic modifications, and early-life developmental outcomes in the offspring. - The review states that the contribution of the paternal microbiome becomes stronger as the child grows older, while the maternal contribution is strongest at birth. What could be the reason for this? Is this simply a result of increased physical contact and shared environment with the father over time, or could there be a more complex biological reason? For example, could it be that the paternal strains are better adapted to the more mature gut environment of an older infant, allowing them to displace the early maternal colonizers?
The increased paternal contribution may be explained by the growing frequency of physical contact and shared environment with the father, as compared to the mother’s predominant influence during pregnancy and early infancy.
In this context, a study by Valles-Colomer et al. analyzed metagenomes from non-cohabiting adult twins who had previously lived together. The authors found that strain sharing between twin pairs significantly decreased with the number of years spent living apart. Notably, monozygotic twins continued to exhibit higher rates of strain sharing decades after cohabitation, compared to dizygotic twins, suggesting a moderate genetic influence beyond the effect of shared living.
These findings indicate that microbial strain sharing in adulthood is more likely attributable to past cohabitation than to long-term parental transmission. Thus, these results may also help explain the growing paternal influence on the child's microbiome as they age. An evolutionary significance could be also hypothesized according to the hologenome theory and holobionts (line 644-656), however this hypothesis is not yet supported by available evidence.
- At the end of the discussion, the authors briefly mention the "evolutionary significance" of paternal inheritance. I find that fascinating. Could the authors elaborate on what the evolutionary advantage of this microbial double inheritance by the parents might be? Does it provide the offspring with a more diverse microbial "toolkit" that prepares them for a broader range of environments, or does it serve as an honest signal of the father's fitness to the next generation?
We thank the reviewer for these valuable suggestions. In the section retitled “Research Gaps and Further Perspectives”, we started from the intervention potential, suggesting future clinical research on preconception modulation of the paternal microbiome through diet or probiotics. We also address the evolutionary context by referencing the hologenome theory and proposing potential adaptive benefits of dual parental microbial inheritance in shaping offspring resilience across generations. We believe that the evolutionary significance of paternal programming may be considered as the selection of more performing microbial strains to better fit to different environments, but nowadays this hypothesis is not yet demonstrated.
- While the review expertly covers the gut, oral and semen microbiome, are there other, less studied paternal microbiomes (e.g. skin, urine) that could also play a significant, albeit smaller, role in shaping the infant's microbiome through direct contact after birth? Is there any preliminary evidence in this area?
Thank you for your thoughtful and insightful question regarding the potential role of skin and urine microbiota in shaping the infant’s microbiome in postnatal period. After conducting a careful review of the current literature, we found that:
- There is no available evidence to support a role of the paternal skin or urinary microbiota in male reproductive health or fertility-related outcomes.
- Microbial transmission through skin-to-skin contact is well recognized, but most of the existing studies focus on mother–infant dyads, with limited data available on father–infant interactions in this context. Specifically, there are no studies evaluating shared microbial strains among paternal urinary or skin microbiota and infant microbiota.
- A recent systematic review by Cordolcini et al. (Behavioral Sciences, 2024) examined paternal tactile behaviors with full-term infants, including skin-to-skin contact (SSC) and spontaneous touch (ST). The review concluded that paternal touch could exert positive biophysiological, behavioral, and psychological effects on both infants and fathers. However, no data was provided regarding the transfer or establishment of specific microbial species as a result of this contact.
Based on this evidence, we have added a note in the “Research Gaps and Further Perspectives” (line 653-656) and “Conclusion” section (line 665-667) of our manuscript to highlight these findings as an important and promising area for future research, particularly regarding the microbial consequences of paternal-infant physical contact beyond reproductive biology.
- Here are minor errors and corrections
- – Line 53: Add full stop after “maternal domain [3,4]”
We changed the structure of the introduction. We checked the full stops at the end of each sentence in the manuscript. - – Line 56-57. The claim that "seminal microbiota could affect implantation and placental development" is not supported by the cited reference [7], which focuses on cytokine signaling in seminal plasma and its effect on sperm quality. There is no discussion of seminal microbiota or their influence on implantation or placental development in that source. I recommend either providing a more appropriate reference or rephrasing the statement to reflect the actual content of reference 7.
Thank you for this clarification. We rewrote the concept to explain the possible role of seminal microbiota on placental development and implantation, as part of seminal plasma and evidence reported by Robertson. - - Line 95: "...Clostridium innocuum, express steroid-metabolizing enzymes..." The comma before "express" should be removed
We removed the comma. - - Line 116: "...Tremellen et al [18]." - A period is missing
We changed the sentence. - - Line 156: "Campylobacter A" - The taxon "Campylobacter A" is unusual.
We confirm that the taxonomic label “Campylobacter A” is reported as such in the original study we cited (Liu et al. available from: https://pmc.ncbi.nlm.nih.gov/articles/PMC9843272/pdf/main.pdf). - - Line 272: "Ambiental exposure" - "Ambiental" should be "Environmental".
We changed with Environmental. - - Page 6, Figure 1: In the "Gut microbiota" box, "insuline resistance" contains a typo. It should be "insulin resistance".
We corrected it. - Page 10, Table 1: "Biffidobacterium" should be corrected to "Bifidobacterium".
We corrected it. - Page 11, Table 1: "animo-modified polystyrene NPs" likely contains a typo and should be "amino-modified...".
We corrected it. - Page 11, Table 1: “Firmicutes” should be written in italic
We corrected it. - Page 12, Line 437: "the paternal contribution stay stable" should be corrected to "stays stable"
We corrected it. - – Ref #80 "programs ẞ 2-cell dysfunction" - Please check the original source (Ng et al., 2010). The correct term is likely "β-cell dysfunction". The "2" seems extraneous.
We corrected it.
Reviewer 4 Report
Comments and Suggestions for Authors
I read with interest the manuscript titled "Paternal microbiota contribution to fetal health".
I want to point out at the outset that the paternal microbiota is increasingly recognized as a factor that potentially influences reproductive success, the development of the offspring's immune system, and long-term health outcomes, which makes the manuscript significant.
The figure must appear at the point of first mention in the text of the manuscript. Is the figure your own or was it taken from another source?
Please focus more on the paternal modulatory influence on the maternal microbiota after fertilization and early embryonic development.
Please elaborate on the animal and partly human studies to date. Human studies are more limited, but suggest potential links between paternal health, microbiota and offspring outcomes. Could modifying paternal microbiota be a strategy to improve fetal and offspring health? Your opinion, please? Please address this in the text of the manuscript.
The discussion is extremely sparse without citing references. I suggest you stick to the structure of a narrative review in which discussion is not necessary. Discussion is more appropriate for systematic reviews, which you have not done.
Section 7 is redundant. Please cite references according to the instructions for authors.
Author Response
I read with interest the manuscript titled "Paternal microbiota contribution to fetal health".
I want to point out at the outset that the paternal microbiota is increasingly recognized as a factor that potentially influences reproductive success, the development of the offspring's immune system, and long-term health outcomes, which makes the manuscript significant.
The figure must appear at the point of first mention in the text of the manuscript. Is the figure your own or was it taken from another source?
Thank you for your contributions. We have moved the figure in the manuscript, as requested (line 90-104). The figure is original and was created by the authors using Power Point.
Please focus more on the paternal modulatory influence on the maternal microbiota after fertilization and early embryonic development.
According to your suggestions, we included a review of the current knowledge on the role of acellular seminal plasma as an introduction to the topic addressed in the section “semen microbiota” (line 190-215). We then described the potential mechanisms proposed by available studies regarding the modulation by the seminal microbiome of signaling molecules, cytokines and extracellular vesicles in maternal reproductive tract (line 242-253).
Please elaborate on the animal and partly human studies to date. Human studies are more limited, but suggest potential links between paternal health, microbiota and offspring outcomes. Could modifying paternal microbiota be a strategy to improve fetal and offspring health? Your opinion, please? Please address this in the text of the manuscript.
Currently, available human studies have demonstrated that paternal inheritance of microbial strains contributes to shaping the early infant gut microbiome (lines 456–478). However, investigations assessing the impact of paternal microbial shifts on the sperm epigenome and fetal health have primarily been conducted in animal models, as detailed in the subsections of the fourth paragraph. In our opinion, both the direct inheritance of microbial strains from the paternal counterpart and the epigenetic mechanisms modulated by microbiota may play a significant role in influencing offspring health. As suggested, we further elaborated on this hypothesis and its potential clinical applications in the “Research Gaps and Further Perspectives” section (lines 636–656) and the “Conclusion” section (lines 673–676).
The discussion is extremely sparse without citing references. I suggest you stick to the structure of a narrative review in which discussion is not necessary. Discussion is more appropriate for systematic reviews, which you have not done.
We followed your suggestion and replaced the “Discussion” section with a newly titled section, “Research Gaps and Further Perspectives,” to better reflect its content and purpose. In this section, we summarized the currently available evidence on mechanisms of transgenerational inheritance, while clearly acknowledging that most data derive from animal studies. Specifically, we emphasized that the impact of gut microbial perturbations on sperm function and embryo development has been demonstrated only in animal models. Furthermore, we have structured and clearly outlined the major directions for future research, including: 1. Expanding microbiome research to less-studied paternal anatomical sites; 2. Investigating interactions between paternal and maternal reproductive tract microbiota; 3. Exploring epigenetic transmission pathways and other potential mediators; 4. Conducting longitudinal human studies to confirm preclinical findings.
Section 7 is redundant. Please cite references according to the instructions for authors.
Thanks for your comment. We rewrote the references according to the instructions provided by the Journal.
Round 2
Reviewer 2 Report
Comments and Suggestions for Authors
Simple summary
The SS was drastically improved. It now has better logical flow, shortened to 205 words. However, terms like gut-germline axis and transgenerational epigenetic inheritance may be challenging for non-specialist readers. Further simplification and shortening would make the ss stronger.
Abstract
The abstract section shows clear improvement, structure and readability. However, here are some comments. Still it is quite dense and long. For example, lines 30-36 show long and compound sentences. Simplify and shorten it. Still abstract is too broad, covering multiple microbiomes. Please, concentrate on key findings and mechanisms of review. What is the most important? Rewrite the final sentence. Provide a strong conclusion.
Introduction
I could see significant improvement in clear structure, broad background, and readability.
Explain the terms “ gut-germline axis”, “ epididymosomes and bacteria-derived extracellular vesicles”.
You mention animal studies without limitations to human data and methodological challenges.
Add a sentence about the relevance of this topic ( the growth of interest to microbiome research, the increase of human infertility).
Male microbiota
The section was improved. Version 2 expands interplay between reproductive health and different types of male microbiota. Improved flow of text, the division into subsections. recent studies were added. However, I have several comments. The section reports conflicting data without discussion why this discrepancy exists. Figure 1 is not discussed in the text.
Environmental factors of male microbiota and male reproductive health
Some sentences are long and compound, which reduces readability.The concepts about oxidative stress and gut dysbiosis and their reproductive impact are repeated. This could be streamlined.
This section, again, reports results of many animal studies without any discussion of their limitations such as relevance to humans.
Table 1 is a great summary of key evidence, however, its summary does not integrate into text properly.
Paternal microbiota effects on offspring health
Fine
Add a couple of sentences analysing study limitations and gaps in human data.
In conclusive sentences summarise future research directions.
Research gaps and further perspectives
You outline the gaps without deep analysis of why these gaps persist.
This section is overloaded with concepts without prioritisation, that may overwhelm the reader.
Conclusion
Limitations are rather listed than critically summarised. Add impactful sentences for researchers and clinicians.
Author Response
Simple summary
The SS was drastically improved. It now has better logical flow, shortened to 205 words. However, terms like gut-germline axis and transgenerational epigenetic inheritance may be challenging for non-specialist readers. Further simplification and shortening would make the ss stronger.
Thank you for your feedback. As you suggested, we deleted the terms “gut-germline axis” and “transgenerational epigenetic inheritance” and rephrased the same concepts in a more accessible way, while maintaining scientific accuracy. We also slightly shortened the Simple Summary to 164 words to enhance readability.
Abstract
The abstract section shows clear improvement, structure and readability. However, here are some comments. Still it is quite dense and long. For example, lines 30-36 show long and compound sentences. Simplify and shorten it. Still abstract is too broad, covering multiple microbiomes. Please, concentrate on key findings and mechanisms of review. What is the most important? Rewrite the final sentence. Provide a strong conclusion.
We have revised the abstract as suggested. Specifically, we reduced its overall length by removing redundant phrasing and focusing on the key findings discussed in the review (gut and seminal microbiota, epigenetic inheritance, and offspring health). We also revised the final sentence to provide a stronger conclusion.
Introduction
I could see significant improvement in clear structure, broad background, and readability.
Explain the terms “ gut-germline axis”, “ epididymosomes and bacteria-derived extracellular vesicles”.
You mention animal studies without limitations to human data and methodological challenges.
Add a sentence about the relevance of this topic ( the growth of interest to microbiome research, the increase of human infertility).
In the revised version of the Introduction, we have added brief explanations for the terms gut–germline axis (lines 70–75), epididymosomes, and bacteria-derived extracellular vesicles (lines 65–68) to improve clarity for readers. We clarified that most available studies are based on animal models, highlighted the lack of human data, and emphasized the potential impact of future research in this area (lines 72–85). Additionally, we noted the growing interest in microbiome research among infertile patients (lines 45–52).
Male microbiota
The section was improved. Version 2 expands interplay between reproductive health and different types of male microbiota. Improved flow of text, the division into subsections. recent studies were added. However, I have several comments. The section reports conflicting data without discussion why this discrepancy exists. Figure 1 is not discussed in the text.
We thank the reviewer for the positive feedback on the revised section. As suggested, we have addressed the conflicting findings in the literature regarding seminal microbial composition by adding a brief discussion of potential reasons for these discrepancies, including differences in sample size, population heterogeneity, methodological variability, and limited species-level resolution (lines 238–247). In addition, we have clarified the relevance of Figure 1 in the main text (lines 89–91), as it summarizes the microbial strains associated with male reproductive health across various anatomical sites, and we have cited Figure 1 throughout the different subsections.
Environmental factors of male microbiota and male reproductive health
Some sentences are long and compound, which reduces readability.The concepts about oxidative stress and gut dysbiosis and their reproductive impact are repeated. This could be streamlined.
This section, again, reports results of many animal studies without any discussion of their limitations such as relevance to humans.
Table 1 is a great summary of key evidence, however, its summary does not integrate into text properly.
According to your suggestions, we have revised this section, simplifying and shortening long sentences. We also streamlined the content to avoid redundancy, particularly regarding the discussion of oxidative stress and gut dysbiosis and their effects on male reproductive health (lines 306–311). To address the concern regarding the predominance of animal studies, we have added a dedicated paragraph acknowledging their limitations and the challenges of translating these findings to humans in the future to acquire clinical relevance (lines 356–366 and lines 409-413). Finally, we have improved the integration of Table 1 into the main text by providing clearer references and transitions that connect the summarized findings to the corresponding discussion.
Paternal microbiota effects on offspring health
Fine
Add a couple of sentences analysing study limitations and gaps in human data.
In conclusive sentences summarise future research directions.
As requested, we have added a brief analysis of the limitations and gaps in the current human data and future research directions in lines 510-518 and lines 547-549.
Research gaps and further perspectives
You outline the gaps without deep analysis of why these gaps persist.
This section is overloaded with concepts without prioritisation, that may overwhelm the reader.
In response, we have revised the section to improve clarity and prioritization of concepts. First, we changed the title of the paragraph to "Main Findings, Research Gaps, and Further Perspectives", as the initial part now outlines the most extensively investigated mechanisms in the current literature. We also relocated the detailed analysis of the Argaw-Denboba study to the "Offspring Growth and Gastrointestinal Disease" section, where it was previously cited, to improve thematic coherence. To avoid overwhelming the reader, we removed certain sentences and references that were less central. Additionally, we restructured the section by first listing the limitations of this narrative review, followed by the methodological limitations of the cited studies. Finally, we concluded with a clear summary of future research priorities.
Conclusion
Limitations are rather listed than critically summarised. Add impactful sentences for researchers and clinicians.
According to your suggestions, we revised the Conclusion highlighting the major translational gap between animal and human research. We emphasized the absence of studies assessing paternal microbiota prior to conception in relation to specific offspring outcomes. To provide impactful guidance for both researchers and clinicians, we underlined the urgent need for well-designed, human studies and standardized clinical trials. Finally, we reinforced the relevance of this line of research for future preconception care strategies.